# Influence of Transformational Leadership Competence on Nurses’ Intent to Stay: Cross-Sectional Study

**DOI:** 10.3390/nursrep15110399

**Published:** 2025-11-12

**Authors:** Norisk Mataganas Adalin, Theresa Guino-o, Bushra Jafer Al Hnaidi, Yousef Alshamlani, Hazel Folloso Adalin, John Paul Ben Silang, Raeed Alanazi, Regie Buenafe Tumala

**Affiliations:** 1Nursing Department, Medical City, King Saud University, Riyadh 12372, Saudi Arabia; nadalin@ksu.edu.sa (N.M.A.); balhnaidi@ksu.edu.sa (B.J.A.H.); yalshamlani@ksu.edu.sa (Y.A.); 2College of Nursing, Silliman University, Dumaguete City 6200, Negros Oriental, Philippines; theresaaguino-o@su.edu.ph; 3Nursing Department, Prince Sultan Military Medical City, Riyadh 12233, Saudi Arabia; hfolloso@psmmc.med.sa; 4Nursing Department, Women’s Wellness and Research Center, Hamad Medical Corporation, Doha 3488, Qatar; jsilang@hamad.qa; 5College of Nursing, King Saud University, Riyadh 12372, Saudi Arabia; raalenazi@ksu.edu.sa

**Keywords:** intention to stay, leadership competency, nurse manager, nursing management, nursing retention, registered nurse, transformational leadership

## Abstract

**Background/Objective:** Transformational leadership (TL) by nurse managers is a modifiable organizational factor consistently linked to improved staff outcomes. However, evidence from the Arab Gulf region, particularly the Kingdom of Saudi Arabia (KSA), is limited. This study aimed to assess the relationship between nurse managers’ TL and staff nurses’ intent to stay and determine which TL dimensions predict intent to stay. **Methods:** A cross-sectional online survey was conducted among staff nurses at a three-hospital academic medical city in Riyadh, KSA. A total of 523 eligible respondents successfully completed the survey, utilizing probabilistic cluster sampling to guarantee representation from various nursing units within the medical city. Nurse managers’ TL was assessed across five dimensions by using the multifactor leadership questionnaire, and staff nurses’ intention to stay was measured using intent to stay scale. Descriptive statistics summarized the respondents’ demographic profile, nurse managers’ TL and staff nurses’ intent to stay. Normality was evaluated using Shapiro–Wilk and Kolmogorov–Smirnov tests. Relationships were examined using Spearman’s rank correlation, and multivariable ridge regression modeled the predictive contributions of the overall TL and its five dimensions to intent to stay. Results were considered significant if *p* < 0.05. **Results:** Nurse managers’ TL exhibited a positive association with staff nurses’ intention to stay in their current positions (*r* = 0.22, *p* < 0.001). Moreover, every dimension of TL demonstrated a strong positive relationship with the intent to stay (all *p*-values < 0.001). Multivariable ridge regression analysis revealed that the overall TL was a significant predictor of the intent to stay (β = 0.13, *p* < 0.001). **Conclusions:** The findings corroborate prior evidence linking TL to retention intentions. This underscores the practical salience of leadership competencies and attributes of nursing leaders, particularly TL, which recognizes the individual needs of staff nurses. This recognition subsequently fosters retention intentions, cultivates supportive nursing work environment, and enhances overall organizational success.

## 1. Introduction

Nursing retention remains a persistent concern for healthcare systems globally. High turnover and vacancies strain staffing, morale, and continuity. Such pressures have been exacerbated by demography, migration, technology, and recent global events [1,2]. Although the World Health Organization [3] projects some improvement in the global nursing shortfall (5.8 M in 2023 to 4.1 Mby 2030), shortages will likely remain in certain regions. It is anticipated that nurses will persist in migrating from low- and middle-income nations to higher-income countries [4]. In this context, identifying organizational levers that can be modified to retain nurses in current positions remains a strategic and operational priority for health leaders. The cost of replacing a registered nurse is approximately AU$22,000 in Australia and US$56,000 in the United States [5,6]. Higher turnover rates are often associated with an increase in adverse events and a decline in patient experience [6]. Research evidence shows that leadership plays a crucial role and has been repeatedly associated with work environment, engagement, and retention outcomes [7,8].

Transformational leadership (TL) is particularly important in complex, high-demand healthcare settings because it emphasizes vision, purpose, and growth [9]. TL is characterized into four key components: idealized influence, inspirational motivation, intellectual stimulation, and individualized consideration [10,11,12]. The component of Idealized Influence as attributes emphasizes a leader’s ethical example and alignment with core values, demonstrating credibility, integrity, and consistency that foster trust. In terms of behaviors, Idealized Influence concentrates on actions such as fairness, honoring commitments, and occasionally making personal sacrifices for the benefit of the group, which clarifies values in everyday decision-making and enhances followers’ confidence [11,12]. Inspirational Motivation involves articulating a clear and compelling vision, establishing significant goals, and motivating all members to collaborate towards shared objectives [11,12]. Intellectual Stimulation refers to the encouragement of new ideas, challenging outdated beliefs, and fostering an atmosphere where individuals feel secure to innovate and learn. Lastly, Individual Consideration pertains to providing tailored support and feedback, acknowledging each individual’s needs and aspirations, and assisting them in their development and engagement [11,12].

Compared with other leadership styles, which rely on directives, collaboration-contingent reward, and corrective action, TL attempts to increase follower intrinsic motivation and identification with organizational goals [13,14]. In healthcare, TL has been associated with trust, resilience, and adaptability during changes, crises, and recovery, making it a practical approach to shape unit culture and professional growth development in nursing [8,9,12]. Previous studies have highlighted the connection between TL exhibited by nurse managers and various staff outcomes, such as engagement, empowerment, job satisfaction, reduced burnout, and a heightened intention to remain in their roles [14,15,16]. Nurses who perceive their managers as demonstrating TL report greater job satisfaction and a more robust commitment to staying in their positions [15]. Work environment factors associated with leadership, including psychological safety, meaningful work, and supportive feedback, are implicated in staff retention pathways [15]. Previous studies also reported a positive relationship between nurse manager TL and intent to stay [17,18] but with variable effect sizes by setting and sample. Some findings are weak but significant [19].

Building on TL, a key proximal outcome of retention, is nurses’ intent to stay [20]. TL may increase intent to stay through clarifying the purpose, modeling values, and recognizing contributions, which could heighten psychological safety and meaning at work [21,22]. These, in turn, are associated with enhanced job satisfaction and organizational commitment and lessened burnout and moral distress, all of which are well-documented antecedents of intent to stay [23]. Job embeddedness and withdrawn cognitions are reduced when the work environment has a number of features: adequate staffing, autonomy, interprofessional respect and support, and opportunity for development [24,25]. These five particular environment features constitute the pathways by which leadership operates, and when they are present, nurses report increased job embeddedness and decreased withdrawal cognitions [25]. According to Buckley and Sipe [26], leader communication quality, fairness, and support for professional growth are positively associated with intent-to-stay scores, especially in high-acuity units where demand and role stress are highest.

Moreover, the consequences of turnover and vacancies are significant. High turnover can drive up workload, deepen moral distress, and increase burnout, all of which can affect quality of care and outcomes [27]. Repeated hiring and onboarding cycles incur financial and opportunity costs while disrupting teamwork at the organization level [27]. In contrast, retaining nurses fosters continuity of care, the accumulation of implicit knowledge, and enhances interprofessional collaboration, all of which contribute to patient safety and quality [28]. In light of these considerations, it is crucial for nurse managers to possess practical, evidence-based guidance on how to support nurses who wish to remain in their roles, particularly through measures that promote their well-being and the efficiency of the healthcare system.

Several theoretical perspectives can help explain why TL promotes intent to stay [29,30,31]. Grounded in social exchange perspective, transformational leaders cultivate high-quality relationships characterized by trust, support, and reciprocity; in return, nurses may respond with strengthened commitment and willingness to remain [30]. Conservation of resource theory suggests that TL can create resource-rich environments through empowerment, recognition, and developmental opportunities that buffer stress and reduce withdrawal cognitions. Job embeddedness emphasizes the links, fit, and sacrifices that keep employees rooted in their roles; TL may strengthen these ties by aligning values and facilitating meaningful work [25]. Across these lenses, mediating experiences, such as psychological safety, empowerment, and perceived organizational support, provide plausible pathways through which perceived TL shapes intent to stay [31,32].

Although TL of nurse managers is frequently associated with nurses’ intention to remain in their positions, the specific TL behaviors of nurse managers that distinctly impact this intention, as well as the strength of this relationship when accounting for work environment factors across different units, remain ambiguous. Previous research has generally aggregated TL scores, employed various measurement approaches, and yielded inconsistent results [8,14,17,19,33]. The degree to which nurses’ direct perceptions of their immediate nurse managers’ TL influence their intention to stay, independent of workplace factors, is still not fully understood. This study seeks to fill this gap by concentrating on nurses’ perceptions of their managers’ TL competence and examining how these perceptions may predict their intention to remain in their roles. Understanding the factors that contribute to nurse retention is more important than ever, especially in today’s rapidly evolving healthcare environment, characterized by increasing demands and resource constraints. This study is timely and relevant because it seeks to provide evidence-based recommendations for healthcare leaders and policymakers to create supportive work environments that foster nurse retention and enhance patient care quality. This study also contributes to the ongoing discourse on effective leadership in healthcare and its critical role in shaping the future of nursing.

### Aim and Hypotheses of the Study

The aim of this study was to assess the relationship between nurse managers’ TL and staff nurses’ intent to stay, and to determine which dimensions of TL serve as predictors for the intention to stay. Specifically, the following hypotheses were established:

**H1:** 
*The perception of nurse managers’ TL by nurses is correlated with their intention to remain in their current positions.*


**H2:** 
*The perception of nurse managers’ TL by nurses serves as a predictor for their intention to stay.*


## 2. Materials and Methods

### 2.1. Research Design

The study employed a quantitative, cross-sectional, and correlational design.

### 2.2. Sample and Setting

The research was conducted at the three hospitals in King Saud University Medical City (KSUMC) with more than 500 beds in Riyadh, Kingdom of Saudi Arabia (KSA). A probabilistic cluster sampling method was used to ensure representation of respondents from all nursing units. The total number of nurses at KSUMC was approximately 2125. The sample size was determined using Slovin’s formula, with an acceptable margin of error of approximately 3.84%. Consequently, the minimum required sample size was 515. We obtained 523 completed responses from nurses who met the inclusion criteria, which slightly surpassed our target and led to an effective margin of error of about 3.8%.

Eligible respondents included bedside nurses who provided direct patient care, had been employed at KSUMC for a minimum of one year, were at least 18 years old, and did not hold any administrative roles. Chief nursing officer, nursing directors, nurse managers, case managers, nurse educators, quality and infection control nurses, charge nurses, and those who had been assigned to their current unit for less than one year prior to data collection were excluded.

### 2.3. Data Collection

Data collection was conducted over a period of more than a month, from 16 May 2024 to 21 June 2024. The researchers distributed a survey flyer with a QR code to each nursing unit, allowing respondents to complete the online survey, which took approximately 6–10 min. In addition, the survey was distributed via email and WhatsApp, with the assistance of gatekeepers, who were nursing directors at the three hospitals of the medical city. We implemented various safeguards to ensure data quality. These measures included the distribution of single-use survey links by gatekeepers to eligible respondents, the activation of the ‘Limit to one response’ feature in the online survey, and the use of time-stamp verification to identify duplicate entries. Additionally, we incorporated an attention-check question and monitored for surveys completed in an unusually short time. Prior to data analysis, we eliminated any entries that appeared to be duplicates or of low quality, adhering to a defined data-cleaning protocol. The survey consisted of three sections, each assessing a specific variable under study, including demographic profile, TL, and intent to stay.

### 2.4. Instruments

This study utilized two validated instruments to measure the variables of interest. The multifactor leadership questionnaire (MLQ), comprising 20 items, was used to evaluate TL, with a high reliability index (Cronbach’s alpha of 0.96) [10]. The tool has five subscales: Idealized Influence—Attributes, Idealized Influence—Behaviors, Inspirational Motivation, Intellectual Stimulation, and Individual Consideration. Each subscale has four items. Each item received a rating ranging from 0 (not at all) to 4 (frequently, if not always). The TL results were compared with the general population’s normative percentile table. In the current study, high reliability was also established with the Cronbach’s alpha values of the five subscales: Idealized Influence—Attributes (0.973), Idealized Influence—Behaviors (0.975), Inspirational Motivation (0.973), Intellectual Stimulation (0.974), and Individual Consideration (0.972), and overall TL (0.965).

The second tool utilized in this study is the intent to stay questionnaire, which was adapted by Iverson [34]. This instrument was employed to evaluate nurses’ intentions to continue their employment at their current institution. The questionnaire consists of four items, each rated on a five-point Likert scale, where each point signifies a distinct level of agreement. Specifically, a score of 1 reflects strong agreement, a score of 2 indicates agreement, a score of 3 represents a neutral stance, a score of 4 signifies disagreement, and a score of 5 denotes strong disagreement. Items 1, 2, and 4 were reverse-coded. Subsequently, higher mean scores, closer to five on the scale, suggest a greater likelihood of the intent to remain within the organization. The scoring and interpretation for each level of the scale delineate the corresponding ranges: 1.00–1.80 indicates very low intent, 1.81–2.60 signifies low intent, 2.61–3.40 reflects average intent, 3.41–4.20 denotes high intent, and 4.21–5.00 represents very high intent. The overall score for a respondent on the scale was calculated by averaging the ratings assigned to each individual item. The instrument’s Cronbach’s alpha was reported to be 0.79 in prior research, indicating a high reliability index for the scale [34]. In the present study’s sample, the Cronbach’s alpha was 0.860.

### 2.5. Statistical Analysis

Statistical analysis was conducted using IBM-SPSS version 30. Descriptive statistics, including frequency counts, percentages, means, and standard deviations, were calculated to provide a comprehensive overview of the demographic profile, the perceived use of TL by nurse managers, and the staff nurses’ level of intent to stay. Spearman’s rank correlation was employed to examine the relationships between TL and with intent to stay, providing insights into the strength and direction of these relationships. It was used because normality of the collected data was not met, as indicated by Shapiro–Wilk test (*p* < 0.001) and Kolmogorov–Smirnov test (*p* < 0.001). Additionally, multivariable ridge regression was used to explore the predictive relationship between the five dimensions of TL and nurses’ intent to stay. Ridge regression is a type of linear regression that mitigates issues of multicollinearity and overfitting by reducing the magnitude of the coefficients [35]. This technique operates by adding a penalty into the standard error minimization process, which relies on the squared values of the coefficients. Consequently, larger coefficients become less favorable, resulting in a model that possesses smaller, more stable weights, which are expected to perform better on new data [35]. Therefore, this approach was chosen due to multicollinearity among the leadership dimensions, as indicated by variance inflation factor values exceeding 10. The level of significance of the study’s findings was established at *p* < 0.05.

### 2.6. Ethical Considerations

Ethics approval was obtained from the Institutional Review Board of KSUMC, Riyadh, KSA (Reference No. E-24-8803, dated 16 May 2024). All procedures conformed to the principles of the Declaration of Helsinki. The respondents received comprehensive study information at the outset of the online survey, including the purpose, potential benefits, foreseeable risks, voluntary nature of participation, and right to withdraw at any time without penalty, and no incentives were offered. Privacy and confidentiality were upheld throughout. Informed consent was obtained via online prior to starting the online survey. Data were anonymized at collection, with responses de-identified and assigned unique codes. No personally identifying information was retained in the analytic dataset. Electronic data were stored in a secure, password-protected OneDrive repository with access solely restricted to the researchers.

## 3. Results

### 3.1. Respondents’ Profile

Table 1 presents the profile of the respondents, comprising 523 registered nurses. The majority of nurses consisted of females (86.23%), within the age range of 20 to 40 years (81.64%), Filipinos (59.27%), and those holding a bachelor’s degree (87.76%). The highest proportion was employed at Hospital A (60.8%), designated at surgical units (20.46%), had over 10 years of professional nursing experience (61.57%), and with 1–3 years of experience in their current unit (41.30%). Furthermore, they reported interacting with their nurse managers 1–3 times every shift (67.11%). Additionally, the majority of nurses worked with female nurse managers (89.87%) and had been working with their current nurse managers for a duration of 1–3 years (68.07%).

### 3.2. Transformational Leadership Competence of Nurse Managers and Nurses’ Intent to Stay

The overall mean of 2.77 was around the 43rd percentile, indicating that 43% of nurses scored lower and 57% of nurses perceived their nurse manager as leaders who have transformational qualities (Table 2). The aggregate mean for Behavior is 2.78, corresponding to the 45th percentile, where 45% of the normed population scored lower, whereas 55% scored higher. The Attributes dimension has an aggregate mean of 2.77, indicating a position around the 43rd percentile, meaning that 43% of the scores are lower and 57% are higher. The Motivation dimension, with an aggregate mean of 2.89, is approximately at the 55th percentile, indicating that 55% of the nurses perceived that their nurse managers had higher motivation than the 45% who scored lower. Stimulation, with a mean of 2.76, is at the 42nd percentile, and Consideration, with a mean of 2.68, is at the 35th percentile, indicating that 35–42% of the nurses perceived lower Intellectual Stimulation and Individual Consideration among their nurse managers. Nurses perceived their nurse managers as having higher TL qualities than the general population in all dimensions except Individual Consideration. Meanwhile, the average intent to stay among nurses has a mean of 3.08, with a standard deviation of 0.88, indicating a moderate level of commitment to their current roles.

### 3.3. Relationship Between Nurse Managers’ Transformational Leadership and Nurses’ Intent to Stay

Table 3 highlights the significantly positive relationship between the TL competence of nurse managers and nurses’ intent to stay. Each domain of TL exhibited a significant positive relationship with nurses’ intention to remain in their positions. Idealized Attributes (*r* = 0.21, *p* < 0.001), Idealized Behavior (*r* = 0.22, *p* < 0.001), Individual Consideration (*r* = 0.20, *p* < 0.001), Inspirational Motivation (*r* = 0.19, *p* < 0.001), and Intellectual Stimulation (*r* = 0.21, *p* < 0.001) exhibited significant positive relationships with the intent to stay. Overall, TL of nurse managers was significantly and positively associated with nurses’ intent to stay (*r* = 0.22, *p* < 0.001).

### 3.4. Multivariable Ridge Regression Analysis of Predictors of Nurses’ Intent to Stay

In this study, TL of nurse managers had significant predictive effects on nurses’ intent to stay based on the results of multivariable ridge regression analysis (Table 4). The overall score of TL emerged as a significant predictor, with an unstandardized coefficient of 0.131 (*p* < 0.001), indicating a strong positive relationship with intent to stay. However, among the individual dimensions, none showed significant predictive power.

## 4. Discussion

The findings underscore the importance of TL in the nursing profession, particularly in sustaining the nursing workforce and enhancing organizational leadership. The majority of nurses viewed their nurse managers as demonstrating TL competence. Specifically, the components of Inspirational Motivation, Idealized Influence (Behavior), Idealized Influence (Attributed), and Intellectual Stimulation were predominantly perceived by nurses as characteristics of their managers, while a few of them noted a relatively low level of Individual Consideration among their nurse managers. These findings align with an integrative review indicating that TL is the most acknowledged and positively regarded style among nurse leaders, emphasizing that inspirational motivation and idealized influence are the most significant traits in clinical environments [36]. A prior study also identified TL as the highest-rated style associated with enhancements in nurse and patient outcomes, with most TL competencies observed at elevated levels, with the exception of Individualized Consideration [37]. The findings could be attributed to the present study’s organization being in the stage of its Magnet journey, during which TL is still being fully institutionalized. The American Nurses Credentialing Center (ANCC) Magnet Recognition Program [38] highlights the importance of nurse managers embodying TL qualities, which significantly impacts the quality of care and organizational effectiveness. This leadership style is integral to the Magnet framework, which emphasizes the development of nursing practice through components like exemplary professional practice and structural empowerment [39]. The ANCC Magnet emphasizes the necessity for organizations to implement mentorship and succession planning programs under the TL standard, providing a structured method to cultivate TL qualities among emerging nurse leaders [40]. Several studies found that mentorship and leadership training programs improved the TL qualities of nurse managers, thus promoting an empowered and accountable culture among nursing staff [33,39]. These results suggest that policymakers need to establish mentorship and succession planning as standard practices that align with the ANCC Magnet TL requirements and are connected to the performance dashboard and leader evaluation.

Moreover, among the TL competencies, inspirational motivation was the most prevalent among nurse managers, which is similar to previous studies in Italy [41], Ethiopia [42], Jordan [16], and the USA [43] but in contrast with the previous studies conducted in Qatar [37] and the Netherlands [44]. The emphasis on inspirational motivation in TL competence among nurse managers in the KSA can be attributed to the diverse cultural backgrounds of expatriate nurses. Nurse managers utilized motivational strategies to bridge cultural differences by emphasizing shared values and collective goals, thereby fostering inclusivity and collaboration. These findings are in agreement with a recent study underscoring the importance of cultural intelligence among nurse managers as a prerequisite for effectively navigating cultural differences and demonstrating TL competence to empower and motivate their workforce [45]. The findings suggest a necessity for leadership programs that leverage the advantages of Inspirational Motivation while simultaneously tackling the shortcomings of Individualized Consideration. This can be achieved through a structured approach to enhancing coaching skills, feedback literacy, and mentoring and sponsorship capabilities. The curriculum of such programs should encompass training in cultural intelligence, scenario-based activities tailored for multicultural teams, and targeted micro-assessments that deliver specific feedback aligned with the dimensions of TL.

Interestingly, the study reported that nurses exhibited an average intention to remain at their current workplace. The result is also similar to a previous study conducted in the Philippines [46], in which a significant portion of nurses had an average intent to stay in their current workplace, indicating a sense of ambivalence towards their intention to remain. The finding also corroborates earlier research carried out in the KSA and South Korea, which indicates a moderate level of turnover intention among nurses [46,47,48]. Conversely, a multicenter study conducted in China showed low intent to stay [49], whereas another study in Ghana reported a higher intent to stay [19]. The present study’s finding could be attributed to individual grit, working conditions, economic factors, and external circumstances, which are similar to previous studies [18,46]. Nurses’ intention to stay in their positions is increasingly being associated with individual characteristics, such as grit, defined as perseverance and passion for long-term goals [50]. The construct of grit may affect nurses’ resilience and satisfaction with their job, which ultimately may influence their desire to remain in their position. Additionally, numerous studies support that nurses intend to stay in their position when they receive adequate salary; excessive career advancement, such as being certified in a nursing specialty; favorable work setting; and good leadership [25,41,51,52]. These results advocate for the adoption of retention strategies at the unit level, which encompass regular scheduling, supportive oversight, and organized mentorship, in addition to recognition opportunities like support for specialty certification. Approaches that foster resilience, such as debriefing and peer coaching, can assist in transforming nurses’ perseverance into sustained success and dedication to the organization.

Another significant result of this study is the positive relationship between the overall perceived TL competence of nurse managers and nurses’ intent to stay, which aligns with theory and previous studies indicating that TL fosters supportive, meaningful, and empowering work climates that enhance commitment and increase intent to stay [12,18,53,54]. These findings may be attributed to the academic tertiary-care context of the present study’s settings, wherein TL competencies amplify access to professional development opportunities; strengthen value congruence with a mission centered on excellence, learning, and innovation; and cultivate a supportive psychosocial climate that mitigates burnout. In such settings, TL can translate institutional resources into individualized growth pathways, enhance psychological empowerment and engagement, and reinforce professional identity, collectively fostering organizational commitment and reducing turnover intentions [55,56]. However, previous studies showed that beneficial relationship may be diminished in highly limited resources or high-stress environments [52,56]. TL expectations without corresponding structural support (adequate staffing, manageable workloads, and psychological safety) may unintentionally increase stress and cynicism, possibly decreasing retention [52,56]. This may imply that policymakers need to ensure healthcare systems implement competency-based TL development for nurse managers, while also integrating structural support such as safe staffing, manageable workloads, and psychological safety. These measures are essential for sustaining benefits, even in high-stress and resource-constrained healthcare environments.

Furthermore, there was a significant positive relationship across subdimensions of TL with retention. Existing research has demonstrated that nurse managers who possess TL skills lead to reduced intentions to leave, through relational and empowering behaviors that foster a positive work atmosphere and strengthen organizational commitment [19,33]. Furthermore, a significant positive relationship was found across subdimensions, suggesting multiple links that promote retention. For instance, idealized attributes and idealized behavior, often combined as idealized influence in TL, are positively related to nurses’ intent to stay because the leaders address core drivers of retention in nursing: trust, respect, meaning, safety, and professional growth. Nurse leaders who embody integrity, consistency, and a strong moral or ethical framework significantly influence the retention and well-being of nursing staff [57]. The findings corroborate previous studies showing that ethical leadership is linked to increased psychological empowerment and organizational justice, which are vital components that influence nurses’ workplace experiences and decisions to stay in their roles [58,59,60]. In turn, these positive experiences promote a sense of belonging, leading to increased staff retention [42]. These findings indicate that implementing ethical leadership development and training focused on empowerment for nurse managers may improve the impact of TL on retention. This improvement is achieved by enhancing perceptions of organizational justice and empowerment, fostering a sense of belonging, and reducing the likelihood of burnout among nurses.

Similarly, the significant positive correlation between intellectual stimulation, individualized consideration, and nurses’ intention to remain in their positions aligns with research indicating that various aspects of TL exert different influences on retention [61,62]. Previous studies have recognized intellectual stimulation and individualized consideration as critical factors that boost creativity, promote professional growth, provide supportive coaching, and foster autonomy. The implications of this finding may be linked to the organizational environment, which is marked by a diverse, multinational workforce and a mission that includes clinical service, education, and research, thereby emphasizing the importance of personalized support. Transformational leaders who tailor coaching, communication, and recognition to individual preferences bridge cultural gaps, strengthen belonging, and reduce turnover among expatriate and local staff alike [20]. By implementing inclusive practices and recognizing the unique backgrounds of each team member, leaders can foster a robust sense of belonging. In addition, leaders who provide individualized development plans (certifications, specialty rotations, research participation, and preceptorship roles) help nurses see clear futures inside the organization rather than seeking advancement elsewhere [25,26,51,52].

This study also indicated a significant positive correlation between Inspirational motivation and the intention of nurses to remain in their positions. This finding supports the results of a previous study, which suggests that when nurses perceive empowerment and support in their roles through effective communication and opportunities to lead initiatives provided by nurse managers, their commitment to staying in the healthcare environment significantly increases [63]. This outcome may stem from the manner in which nurse managers present opportunities for change and the implementation of new practices and technologies in a supportive context within academic medical city. The advocacy for new practices and technologies by nurse managers can improve the overall work atmosphere, which is a vital element affecting the intention to stay [64,65]. Inspirational leaders play a crucial role in enhancing care quality by promoting and supporting evidence-based practice initiatives through ongoing performance evaluation, recognizing areas for improvement, and strategically aligning professional development [56,63]. These results carry practical and beneficial implications for nursing and healthcare management, highlighting the importance of cultivating an environment that encourages TL to strengthen nurse commitment [64,66].

Lastly, the results indicated that only the overall TL score, rather than its individual subdimensions, significantly predicted nurses’ intent to stay. TL is frequently conceptualized as a holistic construct whose primary impact derives from the synergistic interplay of idealized influence, inspirational motivation, intellectual stimulation, and individualized consideration, rather than from any single behavior in isolation. This finding is similar to those of previous studies, which reported that TL exerts a more substantial influence on nurse retention as a holistic construct rather than its discrete elements [20,33,67]. Recent integrative and empirical studies indicated that the protective effects of TL on employee turnover intentions are significantly enhanced through mechanisms such as work engagement and perceived organizational support [20,67]. This finding suggests the importance of coherent, organization-wide leadership practices over fragmented, behavior-focused interventions that may foster a supportive organizational environment that mitigates turnover intentions among nurses. In real practice, these findings highlight the organizational alignment and integrated leadership development through policies, resources, and role modeling as TL as a cohesive package to support nurses’ intent to stay [68,69].

### Limitations of the Study

This study has limitations to consider that hinder the generalizability of the findings. First, the scales were measured via self-report, introducing potential common method variance, recall bias, and social desirability bias. The method of data collection, which was voluntary and conducted online, may have resulted in self-selection bias, as nurses who held more pronounced opinions or had easier access to the survey were more inclined to participate. Second, cross-sectional design limits the ability to establish causal relationships. Third, the intention to remain is a proximal attitudinal outcome and may not necessarily lead to actual retention behavior; longitudinal turnover was not observed. Fourth, while penalized regression effectively addressed multicollinearity among leadership dimensions, it did not consider all potential confounding variables (such as years of experience, organizational size, and unit type/mix). Furthermore, the model accounted for only a modest proportion of variance, indicating that significant unmeasured factors (including staffing ratios, workload acuity, compensation, benefits, team dynamics, and organizational change) are likely to influence retention intentions. Fifth, the sample was drawn from a specific organizational and cultural context with a predominantly female workforce and overrepresentation of particular nationalities. As a result, the findings may not be generalized to other healthcare systems, workforce compositions or geographic regions in the KSA.

## 5. Conclusions

The findings indicated that TL behaviors exhibited by nurse managers—articulating a compelling vision, providing individualized support, fostering intellectual stimulation, and modeling ethical practice—are positively associated with nurses’ intent to stay. Across analyses, higher perceived TL competence corresponded with stronger retention intentions. These results suggest that retention is influenced not only by structural elements and economic circumstances, including staffing and workload, but also by the quality of leadership present in the immediate work environment such as TL competencies of nurse managers. Embedding systematic assessment of nurse manager competency into routine leadership development and workforce planning may be warranted. Integrating regular, evidence-based competency assessment for nurse managers by using validated tools, multisource feedback, and unit performance indicators. These measures can guide targeted development, and align leadership capacity with retention priorities, thereby enhancing the quality of practice environment, and supporting nursing workforce stability.

## 6. Recommendations

Based on the current study findings, nurses who perceive TL as a significant factor are more inclined to remain in their positions. Therefore, we propose the implementation of a targeted leadership enhancement program for nurse managers that specifically focuses on the TL competencies most closely associated with retention. Nursing administration ought to establish structured, ongoing competency evaluations for nurse managers, utilizing validated leadership metrics, 360-degree feedback, and unit-level indicators. This approach will facilitate targeted development, coaching, and succession planning that aligns with workforce stability. Concurrently, policymakers should establish clear competency standards, conduct periodic revalidations, and incentivize leadership competencies and retention through accreditation such as Magnet Recognition, and funding linked to operational benchmarks. Nursing education should embed competency-based leadership training across curricula (transformational behaviors, communication, staffing science, quality/safety, and analytics) with applied practice and mentorship. Future research should use multisite, longitudinal, or quasi-experimental designs to test leadership interventions, probe mechanisms and moderators, and include objective outcomes (turnover, vacancies, and quality/safety) to strengthen inference and generalizability.

## 7. Implications for Nursing Management and Practice

The findings of the present study advocate for the integration of TL competencies in the development of nursing management, highlighting the importance of vision-setting, individual consideration, ethical role modeling, and continuous feedback. Connecting leadership training to retention metrics at the unit level offers guidance in coaching and mentorship aimed at achieving quantifiable workforce results. Hospital policies should incorporate standards for leadership quality in governance, accreditation such as the ANCC Magnet Designation, and performance assessments, which should be directly aligned with the Vision 2030 objectives for a robust and efficient Saudi health workforce. Recognition strategies that align with Saudi cultural values, such as awards from senior nursing executives and non-monetary tokens of appreciation, can foster a sense of belonging and pride among staff nurses. These approaches, in conjunction with well-defined advancement opportunities and supportive staffing policies, may contribute to retaining nursing personnel and reducing turnover.

## Figures and Tables

**Table 1 nursrep-15-00399-t001:** Respondents’ Profile (n = 523).

Profile Variables	*f*	*%*
Gender		
Male	72	13.77
Female	451	86.23
Age (year): Mean = 35.90, SD = 6.73, Range = 22–60		
20–40 years old	427	81.64
41–60 years old	96	18.35
Nationality		
Saudi	45	8.60
Filipino	310	59.27
Indian	146	27.92
Others (Jordanian, Pakistani, African, and Egyptian)	22	4.21
Highest Education Attained		
Diploma in Nursing	55	10.52
Bachelor’s Degree	459	87.76
Post-graduate Degree (Masteral and Doctoral)	9	1.72
Hospitals		
Hospital A	318	60.80
Hospital B	141	26.96
Hospital C	64	12.24
Nursing Unit		
Intensive Care Units	67	12.81
Medical Units	58	11.09
Surgical Units	107	20.46
Out Patient Department Units	88	16.83
Dental Clinics/Units	59	11.28
Perioperative Services	72	13.77
Procedural Units	34	6.50
Emergency Units	38	7.27
Years of Experience as Registered Nurse (year): Mean = 12.01, SD = 6.24, Range = 1–32		
1–3 years	32	6.12
4–6 years	68	13
7–9 years	101	19.31
10 years and more	322	61.57
Years as Registered Nurse in the Unit (year): Mean = 6.40, SD = 5.28, Range = 1–32		
1–3 years	216	41.30
4–6 years	100	19.12
7–9 years	81	15.49
10 years and more	126	24.09
Number of Interactions every shift with Nurse Managers (year): Mean = 2.98, SD = 1.81, Range = 0–9		
No interaction	3	0.57
1–3 times	351	67.11
4–6 times	141	26.96
7–9 times	28	5.35
Gender of Nurse Managers		
Male	53	10.13
Female	470	89.87
Years Working with the Current Nurse Managers (year): Mean = 3.55, SD = 3.12, Range = 1–22		
1–3 years	356	68.07
4–6 years	97	18.55
7–9 years	46	8.80
10 years or more	24	4.59

Note. *f* = Frequency, % = Percentage.

**Table 2 nursrep-15-00399-t002:** Transformational leadership competence of nurse managers and nurses’ intent to stay (n = 523).

Dimensions of Transformational Leadership	Mean	SD
Idealized Influence (Behavior)	2.78	0.88
Idealized Influence (Attributed)	2.77	0.91
Inspirational Motivation	2.89	0.83
Intellectual Stimulation	2.76	0.91
Individualized Consideration	2.68	0.84
Transformational Leadership Overall Mean Score	2.77	0.88
Intent to Stay Overall Mean Score	3.08	0.88

Note. Transformational Leadership scale uses a 5-point Likert rating (0.0 = Not at all, 4.0 = Frequently, if not always). Possible mean score was given as follows: 0–0.80 (Not at all Perceived), 0.81–1.60 (Perceived Once in a while), 1.61–2.40 (Perceived Sometimes), 2.41–3.2 (Perceived Fairly Often), and 3.21–4 (Perceived Frequently, If not always). Nurses’ Intent to Stay scale uses a 5-point Likert rating (1 = Strong agreement, 5 = strong disagreement). Possible mean scores: 1.00–1.80 (Very low intent), 1.81–2.60 (Low intent), 2.61–3.40 (Average intent), 3.41–4.20 (High intent), and 4.21–5.00 (Very high intent).

**Table 3 nursrep-15-00399-t003:** Relationship between transformational leadership of nurse managers and nurses’ intent to stay (n = 523).

Transformational Leadership (Independent Variable)	Nurses’ Intent to Stay (Dependent Variable)	
	*r*	*p*
Idealized Influence (Attributed)	0.21	<0.001 ***
Idealized Influence (Behavior)	0.22	<0.001 ***
Individualized Consideration	0.20	<0.001 ***
Inspirational Motivation	0.19	<0.001 ***
Intellectual Stimulation	0.21	<0.001 ***
Overall Score of Transformational Leadership	0.22	<0.001 ***

Note. *r*—Spearman’s correlation coefficient; *** Significance level at *p* < 0.001.

**Table 4 nursrep-15-00399-t004:** Multivariable Ridge Regression Analysis of Predictors of Nurses’ Intent to Stay (n = 523).

Transformational Leadership	Dependent Variable—Nurses’ Intent to Stay				
	*B*	*SE*	*β*	*t*	*p*
Idealized Influence (Attributed)	0.05	0.07	0.08	0.71	0.48
Idealized Influence (Behavior)	0.00	0.07	0.00	0.02	0.99
Individualized Consideration	0.03	0.06	0.05	0.49	0.62
Inspirational Motivation	0.04	0.07	0.06	0.56	0.58
Intellectual Stimulation	0.01	0.06	0.02	0.18	0.86
Overall Score of Transformational Leadership	0.13	0.03	0.20	4.12	<0.001 ***

Note. *B*—unstandardized coefficient; *SE*—standard error; *β*—standardized coefficient, *t*—*t*-test value. *** Significance level at *p* < 0.001.

## Data Availability

The raw data supporting the conclusions of this article will be made available by the authors on request.

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
