# Peer review of "Influence of Transformational Leadership Competence on Nurses’ Intent to Stay: Cross-Sectional Study"

_nursrep, 2025, doi:10.3390/nursrep15110399_

Round 1
Reviewer 1 Report
Comments and Suggestions for Authors
The authors report a study examining the relationship between transformational leadership of nurse managers, from the perspective of staff nurses’, and the nurses’ intent to stay using a sample specific to Saudia Arabia. The knowledge gained from this study is useful to nurses, their organizations and to researchers who are interested in how effective leadership can impact nurse retention, particularly at a time when nursing turnover and nursing shortages are an immediate challenge globally. The report is thorough, logically structured and well written for the most part. Aside from a few minor grammar edits that are suggested, the only concern is the data in Table 3 which does not align with the interpretation of correlations found in the abstract and the paragraph preceding the table. Details of suggested edits are noted below.
-page1, lines 31-33 – “Among the TL dimensions, individualized consideration and inspirational motivation demonstrated the strongest positive relationships with intent to stay.” According to Table 3, the correlation coefficients for these 2 dimensions were the lowest. Is the error in the data found in the table?
-page 2, lines48-50 – Minor edit, suggest revising to “Although the World Health Organization (WHO, 2025) projects some improvement in the global nursing shortfall (5.8M in 2023 to 4.1M in 2030), shortages will likely remain…”
-line 55, minor edit, suggest revising to “…to replace a registered nurse with higher turnover associated with more adverse events…”
-line 62, minor edit. Change “describes” to “describe”.
-line 64, minor edit. Change "relies” to “rely”.
-lines 69-70, minor edit. Suggest revising to “Previous studies highlighted the link between nurse manager TL and staff outcomes…”
-line 73, minor edit. Suggest revising to “who perceive their manager as more transformational report higher job…”
-line 81. Minor edit. Change “proximate” to “proximal”.
In the paragraph describing TL (lines 60 onward), it might be more instructive to describe each of the TL dimensions here rather than including these details in the instruments section on page 4. Since the aim of the study is to determine which dimensions are predictive of intent to stay, the description of the dimensions could come earlier in the paper.
-page 3, line117, minor edit. Suggest revising to “extensive studies on TL” or “extensive study of TL”
-page 4, second paragraph. I have used the Bass and Avolio (2004) reference in previous research and have not seen these new labels for the dimensions of TL. Is there another reference for these new labels? Alternatively, the rest of the paper continues to use the former standard labels for each dimension so perhaps it might be more clear to remove the discussion of the new labels.
-line 188. A succinct description of ridge regression was helpful as this form of regression is not seen very frequently in this area of research. It would be useful to provide a reference for these statements. Also, I would suggest that you briefly mention the concept of ‘penalization’ so that statement on page 10 (line 408) is more clear for the reader.
-line 181, minor edit. The version of IBM-SPSS used is missing.
-page 5, line 222. “…nurses reported interacting wit their nurse managers 1-3 times.” Please explain the time frame i.e., is it 1-3 times per week, month, etc.
-Table 1 – Minor edit. The number 18 in the “%” column needs to be indented to be consistent with other data. On page 6, please check the “%” for ‘years as RN on the unit (7-9 years)’. And n=81 does not result in a % of 51.5.
-page 7, Table 2. It would be useful to include the range (0-5) for the variables to aid in interpretation (even though you have stated it in the body of the paper).
-Table 3. There are some inconsistencies in the data reported in Table 3. On line264, it is stated that Intellectual Stimulation had a correlation of r=.219 but the table it is stated as .206. On line 265, the overall TL correlation was r=0.200 but is stated as .219 in the table. Please check the data in the table.
-page 8, lines 28-284. The mean of all TL dimensions ranges from 2.68-2.78 indicating only small differences between each dimension. I think it is overstating the results to suggest that “a few perceived lower IS and IC…” There was only a difference of 0.01 difference between IS and II-A.
-page 9, line 340, minor edit. Suggest revising “study” to “studies”.
-lines 346-47, minor edit. Suggest revising to “Furthermore, there was a significant positive relationship across subdimensions of TL with retention.”
-page 10, lines 364-66. Minor edit. I find the use of the word “this” can be unclear. Suggested revision “By implementing inclusive practices and recognizing the unique back-grounds of each team member, leaders can…”.
-page 11, line 424. “managerial capability” was mentioned in the conclusion but this concept was not measured in this study. Suggest removing this term.
Author Response
Reviewer 1 Comments
Comments and Suggestions for Authors
The authors report a study examining the relationship between transformational leadership of nurse managers, from the perspective of staff nurses’, and the nurses’ intent to stay using a sample specific to Saudia Arabia. The knowledge gained from this study is useful to nurses, their organizations and to researchers who are interested in how effective leadership can impact nurse retention, particularly at a time when nursing turnover and nursing shortages are an immediate challenge globally. The report is thorough, logically structured and well written for the most part. Aside from a few minor grammar edits that are suggested, the only concern is the data in Table 3 which does not align with the interpretation of correlations found in the abstract and the paragraph preceding the table. Details of suggested edits are noted below.
RESPONSE: Thank you very much for your positive and constructive comments. We appreciate your time and effort in reviewing our work.
- -page1, lines 31-33 – “Among the TL dimensions, individualized consideration and inspirational motivation demonstrated the strongest positive relationships with intent to stay.” According to Table 3, the correlation coefficients for these 2 dimensions were the lowest. Is the error in the data found in the table?
RESPONSE: Thank you very much for your keen observation about this part. We visited the SPSS output, double-checked the results, and corrected this both in Abstract (Line/s 33-38) and Table 3 as well as in sub-section 3.3 (Line/s 298-307).
- -page 2, lines48-50 – Minor edit, suggest revising to “Although the World Health Organization (WHO, 2025) projects some improvement in the global nursing shortfall (5.8M in 2023 to 4.1M in 2030), shortages will likely remain…”
RESPONSE: We revised this part based on your comment in Line/s 49-53. Thank you very much for this helpful suggestion.
- -line 55, minor edit, suggest revising to “…to replace a registered nurse with higher turnover associated with more adverse events…”
RESPONSE: We have done the revision beyond to this comment after submitting the revised version of work to English language editing. Please refer to Line/s 55-57. Kindly know that we are more than willing to comply with any additional changes regarding this comment, if necessary.
- -line 62, minor edit. Change “describes” to “describe”.
- -line 64, minor edit. Change "relies” to “rely”.
- -lines 69-70, minor edit. Suggest revising to “Previous studies highlighted the link between nurse manager TL and staff outcomes…”
- -line 73, minor edit. Suggest revising to “who perceive their manager as more transformational report higher job…”
- -line 81. Minor edit. Change “proximate” to “proximal”.
RESPONSE: We have edited these parts (Comments 4-8) and further revisions have been implemented after submitting the revised version of work to English language editing. Specifically, the word proximate was changed to proximal.
- In the paragraph describing TL (lines 60 onward), it might be more instructive to describe each of the TL dimensions here rather than including these details in the instruments section on page 4. Since the aim of the study is to determine which dimensions are predictive of intent to stay, the description of the dimensions could come earlier in the paper.
RESPONSE: We have added the description of TL dimensions and correspondingly added the references. Please refer to Line/s 60-75.
- -page 3, line117, minor edit. Suggest revising to “extensive studies on TL” or “extensive study of TL”
RESPONSE: We have done the revision related to this comment. Further revision has been implemented after submitting the revised version of work to English language editing. Please refer to Line/s 126-135.
- -page 4, second paragraph. I have used the Bass and Avolio (2004) reference in previous research and have not seen these new labels for the dimensions of TL. Is there another reference for these new labels? Alternatively, the rest of the paper continues to use the former standard labels for each dimension so perhaps it might be more clear to remove the discussion of the new labels.
RESPONSE: We have done the revision related to this comment and added the description under introduction second paragraph. We removed the new labels and used the former standard labels for each dimension. Please refer to Line/s 60-75.
- -line 188. A succinct description of ridge regression was helpful as this form of regression is not seen very frequently in this area of research. It would be useful to provide a reference for these statements. Also, I would suggest that you briefly mention the concept of ‘penalization’ so that statement on page 10 (line 408) is more clear for the reader.
RESPONSE: We have added a succinct description of ridge regression and briefly mention the concept of penalization related to this comment. Please refer to Line/s 230-235.
- -line 181, minor edit. The version of IBM-SPSS used is missing.
RESPONSE: We have added the version of IBM-SPSS used in this study. Please refer to Line/s 221.
- -page 5, line 222. “…nurses reported interacting wit their nurse managers 1-3 times.” Please explain the time frame i.e., is it 1-3 times per week, month, etc.
RESPONSE: We have done the revision related to this comment. Please refer to Line/s 264-265.
- -Table 1 – Minor edit. The number 18 in the “%” column needs to be indented to be consistent with other data. On page 6, please check the “%” for ‘years as RN on the unit (7-9 years)’. And n=81 does not result in a % of 51.5.
RESPONSE: We have done the revision related to this comment. And added missing figure in the same table (years as RN on the unit).
- -page 7, Table 2. It would be useful to include the range (0-5) for the variables to aid in interpretation (even though you have stated it in the body of the paper).
RESPONSE: This has been addressed in Table 2.
- -Table 3. There are some inconsistencies in the data reported in Table 3. On line264, it is stated that Intellectual Stimulation had a correlation of r=.219 but the table it is stated as .206. On line 265, the overall TL correlation was r=0.200 but is stated as .219 in the table. Please check the data in the table.
RESPONSE: We have done the revision related to this comment. We followed Table 3 and corrected the results on the paragraph.
- -page 8, lines 28-284. The mean of all TL dimensions ranges from 2.68-2.78 indicating only small differences between each dimension. I think it is overstating the results to suggest that “a few perceived lower IS and IC…” There was only a difference of 0.01 difference between IS and II-A.
RESPONSE: We have done the revision related to this comment. Please refer to Line/s 324-329.
- -page 9, line 340, minor edit. Suggest revising “study” to “studies”.
RESPONSE: We have done the revision related to this comment. Please refer to Line/s 401.
- -lines 346-47, minor edit. Suggest revising to “Furthermore, there was a significant positive relationship across subdimensions of TL with retention.”
RESPONSE: We have done the revision related to this comment in Line/s 410-411.
- -page 10, lines 364-66. Minor edit. I find the use of the word “this” can be unclear. Suggested revision “By implementing inclusive practices and recognizing the unique back-grounds of each team member, leaders can…”.
RESPONSE: We have done the revision related to this comment in Line/s 439-441.
- -page 11, line 424. “managerial capability” was mentioned in the conclusion but this concept was not measured in this study. Suggest removing this term.
RESPONSE: The term “managerial capability” has been removed. We sincerely apologize for this typo and/or wrong choice of term. We have done the revision related to this comment in Line/s 503-506.
Thank you so much for taking the time to review our article submission. We appreciate your careful reading and thoughtful feedback, especially for the many helpful suggestion you provided. Your comments have been invaluable in helping us clarify our arguments and improve the overall quality of the revised version of our manuscript.
Reviewer 2 Report
Comments and Suggestions for Authors
Thank you for the opportunity to read and review this manuscript.
Remove "insights" from the title.
Are keywords mesh terms of PubMed?
The introduction is sometimes repetitive. The rationale is descriptive but lacks a clearly defined knowledge gap. The authors should explicitly state what is unknown in the current literature and how this study fills that gap. Some sentences are not supported by references (see lines 118-119)
The introduction should be condensed to focus on the conceptual link between leadership theory and the intention to stay.
A set of hypotheses or at least specific research questions should be articulated. Even in exploratory designs, stating the expected direction of relationships helps justify the analytical approach.
The convenience sampling and “snowball” email recruitment introduce potential selection bias.
The authors should provide details on inclusion/exclusion criteria, geographic distribution, and potential duplication control (a known limitation in web-based surveys).
The inclusion and exclusion criteria are not declared.
Sample size is missing.
Ethical authorisation should declare the number and the date of approval.
The use of ANOVA for group comparisons assumes normality and homogeneity of variance, which is not guaranteed with Likert-scale data. Given that medians and IQRs are reported, non-parametric tests (e.g., Kruskal–Wallis) may be more appropriate.
Age is a continuous variable to report using mean and SD. The same consideration is valid for the years and number...
It is not clear the reason to include also the nursing manager in the data collection. Does the instrument of Avolio evaluate the perception of the leadership style? If yes, you need to evaluate only the nurse's response and being a head nurse is an exclusion criterion. If not, you have to analyse data considering the head nurse and the related nurses' team as a statistical unit.
Additional limitations should be discussed, such as potential self-selection bias typical of online surveys. Cross-sectional design, preventing causal inference. Contextual bias due to timing (pre-/early-COVID). Lack of control for confounders (e.g., years of experience, organisational size).
Recommendations should reflect your research.
Comments on the Quality of English Language
To improve language by an English native.
Author Response
Reviewer 2 Comments
Comments and Suggestions for Authors
Thank you for the opportunity to read and review this manuscript.
- Remove "insights" from the title.
RESPONSE: The title of this study has been revised. The word “insight” has been removed.
- Are keywords mesh terms of PubMed?
RESPONSE: Thank you for your helpful question regarding the keywords. Our originally submitted keywords were author-selected topical terms, and not all were specified as MeSH headings, particularly ‘intention to stay’ which is not explicitly defined as MeSH term. We would like to respectfully request expert assistance from the honorable reviewer regarding this matter, if possible. Thank you very much.
- The introduction is sometimes repetitive. The rationale is descriptive but lacks a clearly defined knowledge gap. The authors should explicitly state what is unknown in the current literature and how this study fills that gap. Some sentences are not supported by references (see lines 118-119).
RESPONSE: We have done extensive revisions related to your comment. References have been added accordingly. Please refer to Line/s 126-142.
- The introduction should be condensed to focus on the conceptual link between leadership theory and the intention to stay.
RESPONSE: This has been addressed in Line/s 46-125. Kindly know that we are more than willing to comply with any additional changes regarding this comment, if necessary.
- A set of hypotheses or at least specific research questions should be articulated. Even in exploratory designs, stating the expected direction of relationships helps justify the analytical approach.
RESPONSE: This has been added in a separate section, ‘Aim and Hypotheses of the Study’ in Line/s 144-151.
- The convenience sampling and “snowball” email recruitment introduce potential selection bias.
RESPONSE: We have not done any revision related to this comment. We employed a probabilistic cluster sampling design to balance representativeness with operational feasibility. Nurses were embedded within organizational units that serve as natural clusters for access and fieldwork. Please refer to Line/s 159-172 and also pointed this out as one of the limitations (Line/s 478-496).
- The authors should provide details on inclusion/exclusion criteria, geographic distribution, and potential duplication control (a known limitation in web-based surveys).
RESPONSE: Thank you for this helpful suggestion. We have clarified these elements in the manuscript as follows. We used several safeguards to protect data quality. These included single-use survey links sent through official channels, and time-stamp checks to spot repeat entries. We also added an attention-check question and looked for surveys finished too quickly. Before analyzing the data, we removed any entries that seemed duplicated or low-quality, following a set data-cleaning protocol. Please refer to Line/s 174-188.
- The inclusion and exclusion criteria are not declared.
RESPONSE: Thank you for this helpful suggestion. We have added these elements in the manuscript as suggested in Lines/s 167-172.
- Sample size is missing.
RESPONSE: We regret the omission and have now added explicit details on the sample size determination and achieved enrollment. Please refer to Line/s 158-166.
- Ethical authorisation should declare the number and the date of approval.
RESPONSE: Thank you for this helpful suggestion. The Institutional Review Board (IRB) approval number and date are already stated in the manuscript’s Institutional Review Board Statement: Approval No. E-24-8803; Approval Date: 16 May 2024. No changes were made to the study’s ethical considerations or approvals. Please refer to Line/s 245.
- The use of ANOVA for group comparisons assumes normality and homogeneity of variance, which is not guaranteed with Likert-scale data. Given that medians and IQRs are reported, non-parametric tests (e.g., Kruskal–Wallis) may be more appropriate.
RESPONSE: Thank you for raising this important point. In our study, we did not compare groups, so we did not use the Kruskal–Wallis test (more than 2 groups) or Mann-Whitney U test (2 groups). Our main goal was to examine the link between transformational leadership and intention to stay. Both were measured with Likert-based composites that did not meet normality according to the Shapiro–Wilk test. Because of this, we used Spearman’s rank correlation, which works well for ordinal data that may not be normally distributed and can show monotonic relationships without needing linearity or equal variances. We report the results with Spearman’s ρ and p-values, using α = 0.05. We did not do any revision related to this comment.
- Age is a continuous variable to report using mean and SD. The same consideration is valid for the years and number...
RESPONSE: Usually, age and years of experience are summarized with the mean and standard deviation when the data are close to a normal distribution. In our dataset, though, these variables were not normally distributed and showed clear skew and clustering at common categories, such as 20–40 or 41–60 years, and 1–3 or 4–6 years of experience. This pattern reflects the staffing mix. To better match the data and make the results easier to interpret for workforce planning, we reported age and experience as grouped frequencies and percentages. This method avoids misleading conclusions that can come from using means with skewed data and shows the actual makeup of the workforce, such as early-, mid-, and late-career groups. Moreover, we have updated the Methods to state this a priori rule, recalculated the summaries, and revised the relevant tables to reflect mean ± SD and range as appropriate, while retaining grouped frequencies in an appendix for workforce planning context. This change aligns our reporting with standard practice and improves interpretability. Please refer to Table 1 and in Line/s 250-270.
- It is not clear the reason to include also the nursing manager in the data collection. Does the instrument of Avolio evaluate the perception of the leadership style? If yes, you need to evaluate only the nurse's response and being a head nurse is an exclusion criterion. If not, you have to analyse data considering the head nurse and the related nurses' team as a statistical unit.
RESPONSE: Thank you for your thoughtful comment. In this study, we look at how bedside nurses view their managers’ transformational leadership by using Avolio and Bass’s Multifactor Leadership Questionnaire (MLQ). This tool is designed to gather nurses’ ratings of their nurse managers’ behaviors in transformational leadership. Since our main goal is to see how nurses’ views of their managers’ leadership relate to their intention to stay, we use the individual nurse as the unit of analysis, not the nurse manager. Moreover, we have added in the manuscript that head nurses/nurse managers, staff in education and quality roles, and those with less than one year in their current unit are excluded in this study.
- Additional limitations should be discussed, such as potential self-selection biastypical of online surveys. Cross-sectional design, preventing causal inference. Contextual biasdue to timing (pre-/early-COVID). Lack of control for confounders (e.g., years of experience, organisational size).
RESPONSE: Thank you for the helpful suggestions. We have expanded the limitations to explicitly address. We have added these points to the limitations section and clarified that inferences concern associations rather than causation.
- Recommendations should reflect your research.
RESPONSE: Thank you for noting that recommendations should reflect our research. We have added a recommendation to directly align with our findings linking nurses’ perceptions of their managers’ transformational leadership to intention to stay in the Abstract (Line/s 38-42), and Recommendations section in Line/s 513-529.
- Comments on the Quality of English Language
To improve language by an English native.
RESPONSE: The manuscript has been sent to a professional English-language editing service by a native English editor. We incorporated all copyediting corrections into the revised version of our submission.
We highly appreciate you taking the time to read and review our work and providing us your thoughtful, helpful feedback. Thank you for paying close attention to how the study was done, how clear it was, and how it was presented. Your feedback has helped us improve the paper, make the statistical justification stronger, and make sure that the recommendations are more in line with our findings.
Reviewer 3 Report
Comments and Suggestions for Authors
The study presents a well-constructed and scientifically sound study of transformational leadership and nurses' intention to stay in the Saudi context. The topic is timely, relevant, and aligns well with global nursing workforce priorities. The introduction provides a sufficient theoretical foundation and a strong rationale for the research. The methodology is rigorous, with appropriate use of established instruments (such as the MLQ and Intent to Stay Questionnaires) and clear justification of statistical techniques such as multiple linear regression to address multicollinearity.
Minor areas for improvement include:
Tables and formatting: Tables are useful, but they require consistent formatting to meet MDPI presentation standards:
Ensure column alignment, especially for numerical data (such as means, standard deviations, and p-values).
Indicate decimal places in tables (two digits are recommended).
Include full table titles that explain all abbreviations, test statistics, and significance markers (p, r, β).
We recommend combining smaller descriptive details (such as percentages and counts) into concise, grouped categories for clarity.
For Tables 2 and 3, specify the sample size (N) in the title and explain the meaning of the percentage comparisons.
Ensure that table titles and numbering comply with MDPI guidelines (centering, bolding, and sequential numbering).
Discussion: The discussion is comprehensive and rich in studies, but some sections are too long and include repeated comparisons between similar studies.
It could be improved by:
Organizing the discussion logically—begin by summarizing the main findings, followed by comparisons with previous research, theoretical explanation, and implications.
Avoid repeating previously presented numerical results in the Results section; instead, focus on why and how the results agree with or differ from existing studies.
Condense citations that present similar points into one or two brief references to maintain a flow of narrative.
Add a short subparagraph on practical implications, particularly how the findings can inform leadership training, staff retention strategies, and hospital policy in Saudi Arabia.
Write a clear conclusion that summarizes the findings into a single, important message for nursing leaders and administrators.
Conclusion and recommendations: These sections are strong; However, the implications for policy development and leadership can be summarized more briefly.
Minor Technical Notes:
Check all in-text citations and reference formatting to ensure consistency with APA/MDPI standards (e.g., spaces before years, punctuation, and DOI links).
Ensure consistent tense is used in the "Methods" and "Results" sections (preferably using the past tense).
Ensure that statistical test assumptions (normal distribution, multicollinearity, etc.) are concisely and clearly justified in the "Methods" section.
Comments on the Quality of English Language
The English language is generally clear, but sometimes verbose or repetitive. Long sentences often contain multiple ideas, which obscure the meaning.
Example: “TL may elevate intent to stay through the processes of clarify-82ing purpose, modeling values, and recognizing contributions,…” - this could be shortened and simplified without loss of meaning.
Some phrases employ nonstandard article usage (e.g., "the transformational leadership" instead of "transformational leadership") and inconsistent verb tenses between the past and present.
Some technical terms (e.g., "ridge regression modeled the predictive contributions") could be clarified for a broader international readership.
A careful proofreading by a native academic or professional editor is recommended to improve fluency, grammar, and readability.
Author Response
Reviewer 3 Comments
Comments and Suggestions for Authors
The study presents a well-constructed and scientifically sound study of transformational leadership and nurses' intention to stay in the Saudi context. The topic is timely, relevant, and aligns well with global nursing workforce priorities. The introduction provides a sufficient theoretical foundation and a strong rationale for the research. The methodology is rigorous, with appropriate use of established instruments (such as the MLQ and Intent to Stay Questionnaires) and clear justification of statistical techniques such as multiple linear regression to address multicollinearity.
Minor areas for improvement include:
- Tables and formatting: Tables are useful, but they require consistent formatting to meet MDPI presentation standards:
RESPONSE: Thank you for this observation. We have reviewed MDPI’s table presentation standards and revised all tables to ensure consistency and compliance.
- Ensure column alignment, especially for numerical data (such as means, standard deviations, and p-values).
RESPONSE: Thank you for the helpful comment. We have reviewed all tables and ensured consistent right alignment for numerical columns (means, standard deviations, test statistics, and p-values), with left alignment reserved for text labels. Numeric formatting has been standardized across tables: means and SDs reported to two decimals, test statistics to two decimals, and p-values reported to three decimals with values below 0.001 shown as < 0.001. We also harmonized the placement of footnotes and ensured that columns with mixed content were split so that numeric entries align uniformly. These revisions have been implemented in Tables 1, 2, 3 and 4.
- Indicate decimal places in tables (two digits are recommended).
RESPONSE: Thank you for the note. We have updated all tables to display numeric values with two decimal places, including means, standard deviations, and test statistics. However, p-values are indicated in three digits. We ensured consistency of presenting the values throughout the Results section.
- Include full table titles that explain all abbreviations, test statistics, and significance markers (p, r, β).
RESPONSE: Thank you for the guidance. We have revised all table titles and legends to include complete explanations of abbreviations and to define all reported statistics and significance markers. Specifically, each table title now states the construct and sample, and footnotes define abbreviations.
- We recommend combining smaller descriptive details (such as percentages and counts) into concise, grouped categories for clarity.
RESPONSE: Thank you for the suggestion. We have reviewed all descriptive tables and combined smaller percentages and counts into concise, grouped categories to improve clarity and readability. Specifically, we collapsed narrow bands (e.g., nursing Unit) into meaningful ranges, merged categories where cell sizes were very small, and reflected these changes in revised table titles. Narrative presentation of Table 1 (Line/s 258-267) has been shortened.
- For Tables 2 and 3, specify the sample size (N) in the title and explain the meaning of the percentage comparisons.
RESPONSE: Thank you for the suggestion. We have updated Tables 2 and 3 to include the sample size (N) directly in each table title (e.g., “Table 2. Respondents Profile (N = 523)”).
- Ensure that table titles and numbering comply with MDPI guidelines (centering, bolding, and sequential numbering).
RESPONSE: Thank you for this observation. We have reviewed MDPI’s table presentation standards and revised all tables to ensure consistency and compliance.
- Discussion: The discussion is comprehensive and rich in studies, but some sections are too long and include repeated comparisons between similar studies.
RESPONSE: Thank you for the constructive feedback. We have substantially edited the Discussion to reduce length, remove repetitive comparisons, and focus on the most relevant contrasts. Please refer to Line/s 322-476. Kindly know that we are more than willing to comply with any additional changes regarding this comment, if necessary.
It could be improved by:
- Organizing the discussion logically—begin by summarizing the main findings, followed by comparisons with previous research, theoretical explanation, and implications.
RESPONSE: Thank you for the helpful guidance. We have reorganized the Discussion to follow the requested sequence: (1) a concise summary of the main findings, (2) comparisons with prior research, (3) theoretical explanation/mechanisms, and (4) practice/policy/research implications.
- Avoid repeating previously presented numerical results in the Results section; instead, focus on why and how the results agree with or differ from existing studies.
RESPONSE: Thank you for this helpful suggestion. We revised the discussion section to minimize repetition of numerical values that are already presented in tables/figures.
- Condense citations that present similar points into one or two brief references to maintain a flow of narrative.
RESPONSE: Thank you for the guidance. We revised the manuscript to condense citations that supported similar points into one or two representative references to improve narrative flow.
- Add a short subparagraph on practical implications, particularly how the findings can inform leadership training, staff retention strategies, and hospital policy in Saudi Arabia.
RESPONSE: Thank you for this valuable suggestion. We added a concise subparagraph on practical implications tailored to the Saudi context. The revised text emphasized that the findings could inform leadership training by prioritizing evidence-based modules on transformational and supportive supervisory behaviors, structured feedback, and recognition practices aligned with Saudi cultural norms and Vision 2030 workforce goals.
- Write a clear conclusion that summarizes the findings into a single, important message for nursing leaders and administrators.
RESPONSE: We addressed this in Line/s 498-511.
- Conclusion and recommendations: These sections are strong; However, the implications for policy development and leadership can be summarized more briefly.
RESPONSE: We added a separate section related to this comment, “Implications for Nursing Management and Practice” in Line/s 531-544.
- Minor Technical Notes:
- Check all in-text citations and reference formatting to ensure consistency with APA/MDPI standards (e.g., spaces before years, punctuation, and DOI links).
RESPONSE: We are confident that the revised version of our work completely adhered to in-text citations and referencing format of the journal, Nursing Reports.
- Ensure consistent tense is used in the "Methods" and "Results" sections (preferably using the past tense).
RESPONSE: This has been addressed throughout the Methods and Results sections.
- Ensure that statistical test assumptions (normal distribution, multicollinearity, etc.) are concisely and clearly justified in the "Methods" section.
RESPONSE: This has been addressed in Line/s 221-238.
Comments on the Quality of English Language
- The English language is generally clear, but sometimes verbose or repetitive. Long sentences often contain multiple ideas, which obscure the meaning.
Example: “TL may elevate intent to stay through the processes of clarify-82ing purpose, modeling values, and recognizing contributions,…” - this could be shortened and simplified without loss of meaning.
RESPONSE: Long sentences have been broken down into shorter, concise and more readable ones to properly and clearly emphasize the study’s flow of ideas. This revision applies to similar instances throughout the revised version of our work. Additionally, the revised version has undergone second round of English language editing before resubmission.
- Some phrases employ nonstandard article usage (e.g., "the transformational leadership" instead of "transformational leadership") and inconsistent verb tenses between the past and present.
RESPONSE: This has been addressed throughout the revised version of our work.
- Some technical terms (e.g., "ridge regression modeled the predictive contributions") could be clarified for a broader international readership.
RESPONSE: This has been clarified in Line/s 230-235.
- A careful proofreading by a native academic or professional editor is recommended to improve fluency, grammar, and readability.
RESPONSE: The revised version of our work has undergone second round of English language editing before resubmission.
We highly appreciate you taking the time to read and review our work and providing us your thoughtful, helpful feedback. Thank you for paying close attention to how the study was done, how clear it was, and how it was presented. Your feedback has helped us improve the paper, make the statistical justification stronger, and make sure that the recommendations are more in line with our findings.
Round 2
Reviewer 2 Report
Comments and Suggestions for Authors
Thank you for addressing and revising your manuscript. I believe that your revisions have significantly enhanced the quality of the manuscript's presentation.